# Ligand-enabled ruthenium-catalyzed *meta*-C−H alkylation of (hetero)aromatic carboxylic acids

Xianglin Luo[1], Peichao Hou[1], Jiayi Shen[1], Yifeng Kuang[1], Fengchao Sun[1], Huanfeng Jiang [1], Lukas J. Gooßen [2]✉ & Liangbin Huang [1]✉

Carboxylates are ideal directing groups because they are widely available, readily cleavable and excellent linchpins for diverse follow-up reactions. However, their use in *meta*-selective C−H functionalizations remains a substantial unmet catalytic challenge. Herein, we report the ruthenium-catalyzed *meta*-C−H alkylation of aromatic carboxylic acids with various functionalized alkyl halides. A bidentate *N*-ligand increases the electron density at the metal center of *ortho*-benzoate ruthenacycles to the extent that single-electron reductions of alkyl halides can take place. The subsequent addition of alkyl radicals is exclusively directed to the position *para* to the $C_{Ar}$−Ru bond, i.e., *meta* to the carboxylate group. The resulting catalytic *meta*-C−H alkylation extends to a wide range of (hetero)aromatic carboxylic acids including benzofused five-membered ring heteroarenes but no pyridine derivatives in combination with secondary/tertiary alkyl halides, including fluorinated derivatives. It also allows site-selective C5−H alkylation of 1-naphthoic acids. The products are shown to be synthetic hubs en route to *meta*-alkylated aryl ketones, nitriles, amides, esters and other functionalized products.

The carboxylate group is a key functionality in many natural products, drugs and functional materials. Aromatic carboxylic acids are widely available in great structural diversity, are easily synthesized from readily available chemicals, can be interconverted into various other functional groups, and can be removed enabling their use as traceless directing group[1]. Due to these advantages, carboxylate groups are highly attractive directing groups for catalytic C−H activation reactions. However, the extension of catalytic concepts from strongly coordinating, often complex nitrogen donors to these simple, weakly coordinating functionalities is extremely challenging[2,3]. In 2007, the Daugulis and Yu group reported pioneering *ortho*-arylations of aromatic carboxylic acids in the presence of a palladium catalyst[4,5]. Since then, extensive research has led to the discovery of various catalytic *ortho*-C−H functionalization of native aromatic carboxylates in the presence of various metal catalysts[2,3]

(Fig. 1a). However, there are no transition-metal catalyzed processes, in which aromatic carboxylate groups direct C−H functionalizations into their *meta*-position.

Several strategies have been devised that direct catalytic C−H functionalizations of other aromatic compounds towards the position *meta* to a functional group[6]. Elegant proof-of-concept studies by the groups of Smith, Hartwig, and others based on steric or electronic control[7–11], the group of Yu and others using template assistance[12–14], the groups of Kuninobu, Kanai, and Phipps using non-covalent interactions[15,16], and the groups of Yu and Dong using transient mediators[17–19], along with Larrosa and other groups using tracelessly removable components[20–24], have demonstrated that *meta*-C−H functionalization can be achieved. However, it has proved difficult to transfer these concepts from complex donor functionalities to native aromatic carboxylic acids.

[1]Key Laboratory of Functional Molecular Engineering of Guangdong Province, School of Chemistry and Chemical Engineering, South China University of Technology, 510641 Guangzhou, China. [2]Ruhr-Universität Bochum Lehrstuhl für Organische Chemie, Universitätsstraße 150, 44801 Bochum, Germany. ✉e-mail: lukas.goossen@ruhr-uni-bochum.de; huanglb@scut.edu.cn

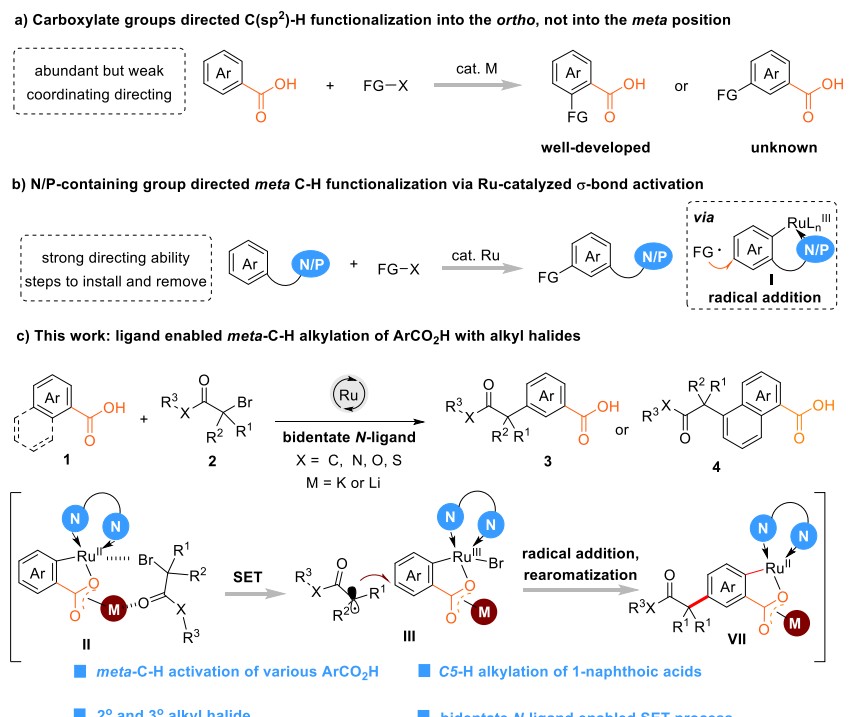

a) Carboxylate groups directed C(sp²)-H functionalization into the *ortho*, not into the *meta* position

b) N/P-containing group directed *meta* C-H functionalization via Ru-catalyzed σ-bond activation

c) This work: ligand enabled *meta*-C-H alkylation of ArCO₂H with alkyl halides

**Fig. 1 | Carboxylate directed *meta*-C–H alkylation as an unmet catalytic challenge. a** Carboxylate groups directed C(sp²)-H functionalization into the *ortho*, not into the *meta* position. **b** N/P-containing group directed *meta*-C–H functionalization via Ru-catalyzed σ-bond activation. **c** This work: ligand-enabled *meta*-C–H alkylation of ArCO₂H with alkyl halides.

A particularly attractive reaction concept has been realized by Ackermann[25–28], Frost[29–31], Liang[32], Zhang[33], and others[34–37] based on ruthenium catalyst (Fig. 1b). In this σ–activation strategy, metal catalysts are directed towards the C–H group *ortho* to nitrogen/phosphine-based donor functionalities with formation of electron-rich ruthenacyles. These intermediates reduce electrophiles to form the corresponding radicals, which then attack the most reactive C–H group of the oxidized ruthenacycles **I**, i.e., the one in *para*-position to the $C_{Ar}$–Ru[III] bond. This allows the installation of alkyl groups *meta* to comparably compact *ortho*-directing groups (Fig. 1b). However, the strategy is so far limited to strongly coordinating, hard to install and remove functionalities such as pyridines[26–37], imines[25,32] and phosphines[38,39]. The use of desirable more ubiquitous functionalities such as carboxylates has not yet been achieved. It poses three substantial challenges: (1) As carboxylates are hard *O*-nucleophiles, they transform only slowly into metallacycles **II**[22,40–43]. This C–H activation step is rate-determining in most *ortho*-C–H functionalizations of carboxylates[22,44]. Instead, an esterification with the alkyl halide coupling partner is a common side reaction. (2) The ruthenacycles formed from aryl carboxylates are less electron-rich than those formed with nitrogen/phosphine-based direction groups. This lowers their redox potential and thus their capability to promote single-electron transfer to the electrophiles. Moreover, the position *para* to the Ru–C bond is less activated towards the addition of alkyl radicals in *ortho*-carboxylate ruthenacycles **III** than in electron-rich pyridine-bearing ruthenacycles (Fig. 1c).

We hypothesized that a catalytic concept for the so far elusive *meta*-C–H alkylation of aromatic carboxylate must address these issues in the following way: 1) The tendency of the alkyl halides to undergo an $S_N2$ reaction with carboxylates must be reduced. 2) Electron-rich ligands must be added to increase the electron density of the resulting ruthenacycles, thus increasing the reduction potential of **Int. II**[45] and the C–H reactivity of **Int. II** (Fig. 1c). 3) Non-covalent interactions between the carboxylate group and functional groups in

the alkyl halides would be beneficial to bring the reactants into closer proximity[46,47], thereby facilitating the single electron reduction and the radical addition steps. We herein report how this concept has led to the successful development of a *meta*-C–H alkylation of aromatic carboxylic acids with various 2°- and 3°-alkyl halides based on a ruthenium-catalyst activated by strongly coordinating, chelating bidentate *N*-ligand.

## Results

### Investigation of reaction conditions

We probed the feasibility of the envisioned reaction concept using the reaction of the amide-functionalized alkyl bromide **1a** with benzoic acid **2a** as the model. As expected, the state-of-the-art catalyst system that is highly efficient in pyridine-directed *meta*-C–H functionalizations ([Ru(*p*-cym)Cl₂]₂ as catalyst, K₂CO₃, Na₂CO₃ or KOAc as base, 1,4-dioxane as solvent) did not give any of the desired *meta*-C−H alkylation product **3aa**[26,27,29,30,33–37] (Table 1, entry 1, for details see Supporting Information, Supplementary Fig. 7). Instead, side reactions of the alkyl halide **1a**, namely elimination (**4a**) with follow-up dimerization (**5a**) or dehalogenation (**6a**) were observed. We next probed whether the activity of the Ru-catalyst could be enhanced by the addition of coordinating ligands and found that bipyridines not only increased the overall conversion but also shifted the selectivity towards the desired product **3aa** (entries 2-7). Best results were obtained with electron-rich bipyridine ligands **L4** and **L6**, whereas electron-withdrawing substituents at the ligands lowered the catalytic efficiency (Table 1, entries 5-7). These results are consistent with our hypothesis that the electron density at the metal center of the ruthenacycle **Int. II** must be increased to facilitate the electron transfer to the electrophile. The solvent properties turned out to be the decisive factor for achieving the desired selectivity for a cross-coupling between **1a** and **2a**. A protic, relatively acidic reaction medium was found to be uniquely effective (entries 8-10). In a 9:1 *ᵗ*BuOH / HFIP solvent mixture, **3aa** was formed almost exclusively, with the previously dominating side products **4a**-

**Table 1 | Optimization of the reaction conditions for *meta*-alkylation of benzoic acid[a]**

| entry | additive | base | ligand | solvent | yield of 3aa [%][b] | byproducts 4a/5a/6a [%][c] |
|---|---|---|---|---|---|---|
| 1 | – | $K_2CO_3$ | – | 1,4-dioxane | n.d. | 17/0/26 |
| 2 | – | $K_2CO_3$ | L1 | 1,4-dioxane | 9 | 17/0/36 |
| 3 | – | $K_2CO_3$ | L2 | 1,4-dioxane | 25 | 6/12/19 |
| 4 | – | $K_2CO_3$ | L3 | 1,4-dioxane | 10 | 7/<5/15 |
| 5 | – | $K_2CO_3$ | L4 | 1,4-dioxane | 27 | 6/9/15 |
| 6 | – | $K_2CO_3$ | L5 | 1,4-dioxane | Trace | 13/0/28 |
| 7 | – | $K_2CO_3$ | L6 | 1,4-dioxane | 31 | <5/12/11 |
| 8 | – | $K_2CO_3$ | L6 | HFIP | n.d. | 23/<5/30 |
| 9 | – | $K_2CO_3$ | L6 | $^t$BuOH | 41 | <5/<5/<5 |
| 10 | – | $K_2CO_3$ | L6 | $^t$BuOH: HFIP = 9:1 | 50 | <5/<5/<5 |
| 11 | – | KOAc | L6 | $^t$BuOH: HFIP = 9:1 | 61 | <5/<5/<5 |
| 12 | – | LiOAc | L6 | $^t$BuOH: HFIP = 9:1 | n.d. | 40/0/46 |
| 13 | – | $Zn(OAc)_2$ | L6 | $^t$BuOH: HFIP = 9:1 | Trace | 10/0/14 |
| 14 | – | $Mg(OAc)_2$ | L6 | $^t$BuOH: HFIP = 9:1 | Trace | 14/0/16 |
| 15 | LiBr | KOAc | L6 | $^t$BuOH: HFIP = 9:1 | 86 (78), 85[d] | <5/0/<5 |
| 16 | LiOAc | KOAc | L6 | $^t$BuOH: HFIP = 9:1 | 84 | <5/0/<5 |
| 17 | AgOTf | KOAc | L6 | $^t$BuOH: HFIP = 9:1 | 53 | <5/<5/<5 |
| 18 | KBr | KOAc | L6 | $^t$BuOH: HFIP = 9:1 | 51 | <5/<5/<5 |
| 19 | $Sc(OTf)_3$ | KOAc | L6 | $^t$BuOH: HFIP = 9:1 | 85 | <5/<5/<5 |

[a]Reaction conditions: **1a** (0.1 mmol), **2a** (0.2 mmol), [Ru(*p*-cym)Cl$_2$]$_2$ (5.0 mol%), ligand (10.0 mol%), additive (30 mol%), base (2.0 equiv.), solvent, 100 °C for 12 h.
[b]Yields of the corresponding methyl esters after esterification with K$_2$CO$_3$ (2 equiv.) and MeI (5 equiv.), and isolated yield in parentheses.
[c]Yields were determined using GC yields with *n*-tetradecane as the internal standard.
[d][Ru(*p*-cym)Cl$_2$]$_2$ (2.5 mol%), 5,5'-di-Me-bpy (5 mol%).

**6a** being observed only in trace quantities. Reducing the pKa of the base led to a step-up in the yields, with best results being obtained with KOAc (entry 11). The presence of a potassium cation is crucial to achieve selectivity for the targeted cross-coupling. With other cations (Li, Zn, Mg), elimination **4a** or dehalogenation products **6a** were the predominant products (entries 12-14). This indicates that solvent-stabilized, potassium-bridged assemblies of the two substrates (**II, III**) are involved in the selectivity-determining steps. The high efficiency of a weak base and a proton-active solvent suggested that the release of product **3aa** from **III** via protodemetalation and salt metathesis with **2a** might still be sluggish. We, thus, added Lewis acids to facilitate these steps. Indeed, the presence of lithium bromide markedly improved the conversion without negatively affecting the selectivity (entries 15-18). This effect can be assigned to the Lewis acidic cation rather than the counter ion, because neither the addition of excess KBr nor the removal of bromide by silver salts has a decisive effect, whereas other Lewis acids such as Sc(OTf)$_3$ were also beneficial (entries 17-19).

## Substrate scope

With the optimized parameters established, we next explored the scope of the *meta*-alkylation with regard to the aromatic carboxylic acids (Fig. 2). Many *ortho*-substituted aromatic carboxylic acid were selectively converted into 1,2,3-trisubstituted arenes. The preference for these thermodynamically less favorable products indicates the sensitivity of the *ortho*-metalation step towards steric hindrance. Alkyl (**3ba-ca**), alkoxy (**3da**), fluoro (**3ea**) and chloro (**3fa**) substituents were all tolerated whereas-in analogy to related processes–nucleophilic hydroxyl and amino groups were found to be incompatible. The directed alkylation of *meta*-substituted aromatic carboxylic acids delivers 1,3,5-trisubstituted aromatic carboxylic acids (**3ga–ia**). Benzoates bearing *para*-substituents were also smoothly converted (**3ja–pa**), which shows that in contrast to the metalation, the radical addition step is not hampered by steric hindrance. Multi-substituted aromatic acids bearing various functional groups were also successfully converted into the corresponding products (**3qa-va**). The reaction extends to benzofused carboxylates and bicyclic heterocycles such as naphthalene (**3wa**), benzodioxane (**3xa-za**), 1-methylindole (**3ab**), benzofuran (**3ac**) and benzothiophene (**3ad**) but not yet to five- or six membered heteroarene carboxylates. Unfortunately, strongly coordinating 6-membered heterocycles such as pyridines or pyrimidines are not tolerated. The scalability of the protocol was demonstrated by a gram scale synthesis of **3ba** (1.2 g, 68%).

We next investigated the scope of the reaction with regard to the alkyl bromide (Fig. 2). The presence of a coordinating functionality in the alkyl bromide was confirmed to be vital. Whereas *t*-butyl bromide gave no conversion, various tertiary α-bromo amides, esters, thioesters and ketones were smoothly transformed into the desired product (**3ae-ax**). Interestingly, even complex α-bromo amides derived from amino acids, such as Gly, Val, Asp, Ser, Phe and Met, all gave good yields (**3ak–ap**). Expectedly, competing esterification could not fully be suppressed for sterically less hindered secondary alkyl bromides (**3ay**), and esters were found as the main products for primary alkyl halides (**3az**). The selectivity for C–C coupling over esterification was found to be particularly high for difluoro alkyl electrophiles (**3bb–mb**), which substantially enhances the preparative utility of the transformation[48]. After all, α-CF$_2$ carbonyl groups are desirable functionalities in drug discovery, as they are stable towards metabolic degradation via enolate mechanisms[49,50]. This moiety is found in the pharmacophores of several commercial drugs, e.g., Tafluprost and Gemcitabine. The *meta*-difluoroalkylation also extends to α-bromo fluorinated esters or amides derived from bioactive molecules such as L-menthol (**3nb**), galactolipin (**3ob**), borneol (**3pb**), estrone (**3qb**), mexiletine (**3rb**) and aminoglutethimide (**3sb**).

Another interesting observation was made when using 1-naphthoic acid as the substrate. In contrast to 2-napthoic acid, the

carboxylate substituent directs the alkylation exclusively into the second aromatic ring, to the C5-position. The regioisomeric identity of the C5-alkylation product was unambiguously confirmed by single crystal X–ray diffraction analysis (Fig. 3). The C2 metalated intermediate **IV**, which should form preferentially, would deactivate the C5 position towards radical addition. Hence, one must assume that the cyclometallation is at least partially directed towards the C8 position. Intrigued by this unusual selectivity pattern, we extended the 1-naphthoic acid reaction variant to various α-bromo amides and esters (**4aa–ua**) including derivatives of borneol (**4oa**), L-menthol (**4pa**), galactolipin (**4qa**), mexiletine (**4ra**), estrone (**4sa**), cholesterol (**4ta**) and tocopherol (**4ua**).

## Mechanistic investigations

As shown in Fig. 4, various control experiments were conducted to shed some light on the reaction mechanism: A) The attempted reaction of 2,6-dimethylbenzoic acid (**2tb**) or aromatic carboxylic acids with two *meta*-methyl substituents (**2ub**) gave no conversion, which confirms that the presence of *ortho*-C–H bonds is vital for the *meta*-functionalization to proceed. B) In the reaction of deuterated substrate **2a-[D$_5$]** with **1a**, substantial D/H scrambling was observed in both *ortho* position at incomplete conversion. This indicates that the *ortho* C–H metalation step takes place rapidly and that it is reversible. C) A negligible kinetic isotope effect value (KIE) of 1.1 was observed when converting a mixture of **2a** and **2a-[D$_5$]**, which suggests that the reversible *ortho*-C–H bond insertion is fast and reversible. Without ligand, the H-D exchange of **2a-[D$_5$]** under standard conditions only gives 65% within 2 h. When **L6** is added, the yield is increased to 95% (for details see Supporting Information), indicating that the bipyridine ligands accelerates the C–H activation step. D) When subjecting **1a** along with a preformed cyclometallated carboxylate complex **Ru-A** to the reaction conditions, no conversion was observed. Only when ligand **L6** was added, the product **3aa** was formed in significant amounts. In combination with **L6, Ru-A** has a comparably high catalytic activity as [RuCl$_2$(*p*-cym)]$_2$. These findings support the intermediacy of *ortho*-benzoate ruthenacycles in the reaction and underline the vital importance of the bipyridine ligand **L6** also for the steps following the *ortho*-metalation. In the reaction using [RuCl$_2$(*p*-cym)]$_2$, GC monitoring of the reaction revealed that *p*-cymene is liberated within the first minutes (for details see Supporting Information, Supplementary Fig. 11), indicating the displacement of this ligand with **L6** during catalyst activation.

E) When conducting the reaction in the presence of the radical scavenger 1,1-diphenylethylene, *meta*-alkylation was retarded, and alkyl radical-capture product was detected. This finding supports the proposed radical mechanism. F) When the rate of the diphenylethylene adduct formation was monitored by in situ GC spectroscopy using a Ru-catalyst with and without ligand **L6**, it was found that ligand **L6** accelerates this reaction. The same observations were made when employing the cyclometallated complex **Ru-A** as a catalyst (see the Supporting Information, Supplementary Table 10). G) Using cyclovoltammetry, the reduction potential of the alkyl halide **1a** was determined as −1.01 V, and that of the [RuCl$_2$(*p*-cym)]$_2$ as −1.12 V. Thus, the [RuCl$_2$(*p*-cym)]$_2$ should already be able to reduce the alkyl halide to **1a**. However, the addition of ligand **L6** should facilitate this step, as it increases the reduction potential to −1.15 V. The preformed pyridine-stabilized, cyclometallated complex **Ru-A** has an even higher reduction potential of −1.22 V. H) When stirring 1-naphthoic acid **2am** with D$_2$O and the ruthenium catalyst, deuterium was incorporated only at the C2 in the absence of **L5**, whereas both C2 and C8 were fully deuterated in the presence of **L5**. This illustrates the crucial importance of the ligand for a successful functionalization in C5 position. I) Based on these control experiments, a plausible mechanism for *meta*-C–H alkylation of aromatic carboxylic acids is proposed. Initially, [Ru(*p*-cym)Cl$_2$]$_2$ reacts with **L6**

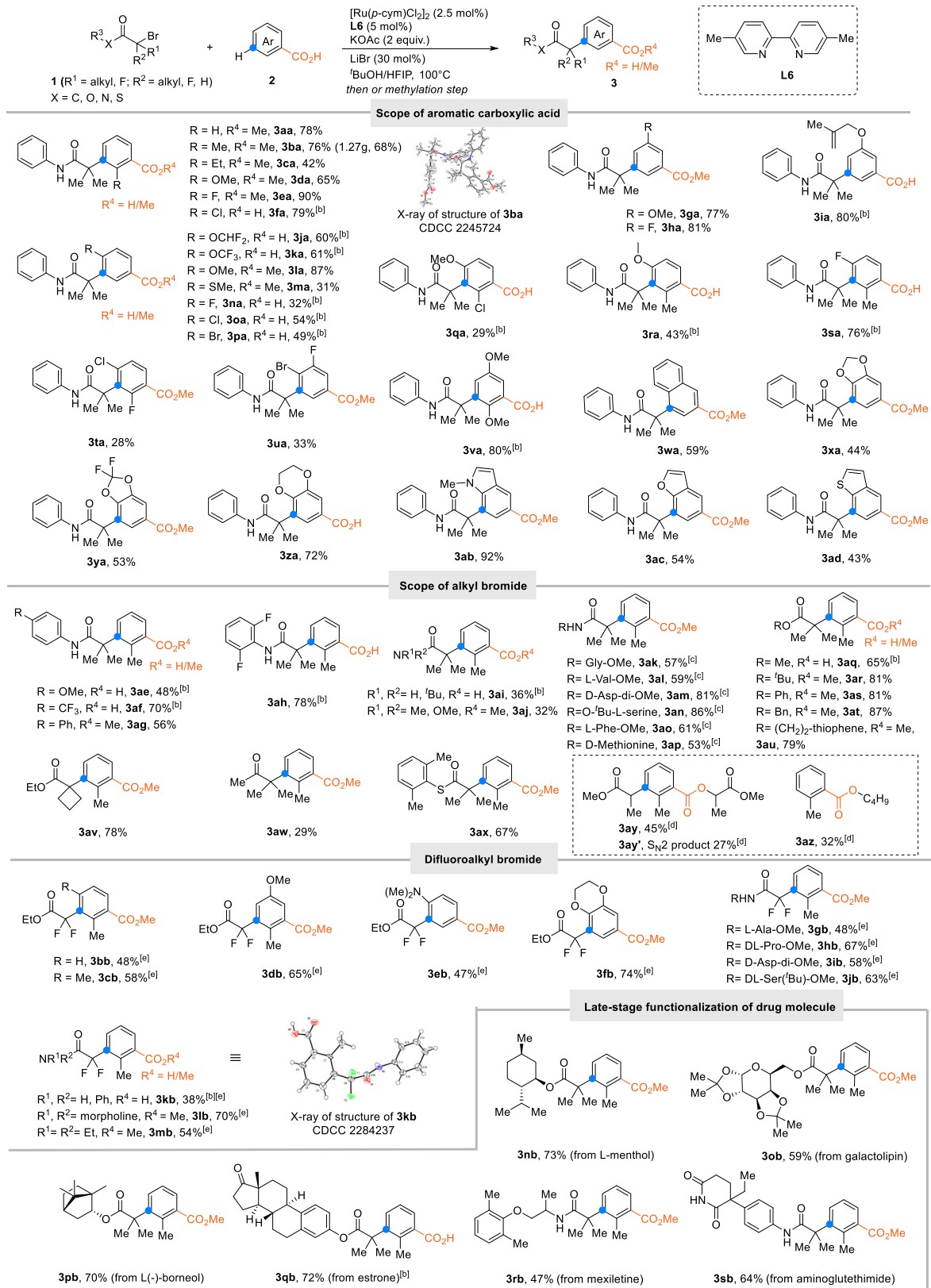

**Fig. 2 | The scope of the *meta*-alkylation of aromatic carboxylic acids.**
[a]Reaction conditions: **1** (0.3 mmol), **2** (0.6 mmol), [Ru(*p*-cym)Cl₂]₂ (2.5 mol%), **L6** (5.0 mol%), KOAc (2 equiv.), LiBr (30 mol%), *t*BuOH/HFIP = 9:1, 100 °C for 12 h under N₂. Yields of the corresponding methyl esters after esterification with K₂CO₃ (2 equiv.) and MeI (5 equiv.) in NMP; [b]Isolated as the free acid; [c]1,4-dioxane instead

of *t*BuOH/HFIP = 9:1; [d]**1** (1.2 mmol), **2** (0.3 mmol), [Ru(*p*-cym)Cl₂]₂ (5.0 mol%), **L5** (10.0 mol%), Na₂CO₃ (2 equiv.), AgOTf (20 mol%), *t*BuOH/HFIP = 20:1; [e][Ru(*p*-cym) Cl₂]₂ (5.0 mol%), **L6** (10.0 mol%), AgOTf (20 mol%) instead of LiBr, *t*BuOH instead of *t*BuOH/HFIP.

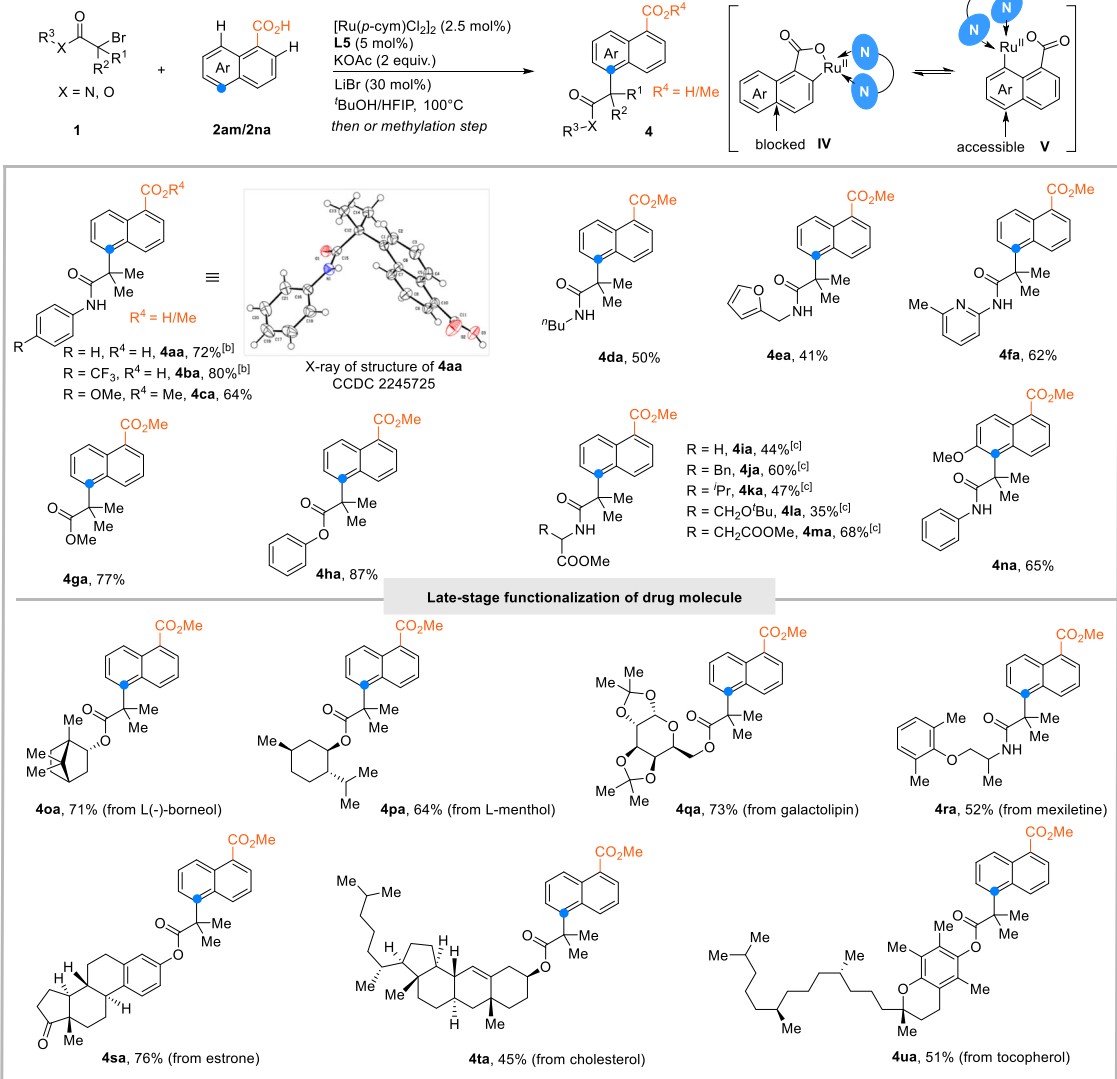

**Fig. 3 | The scope of C5-alkylation of 1-naphthoic acid.** [a]Reaction conditions: **1** (0.3 mmol), **2** (0.6 mmol), [Ru(p-cym)Cl₂]₂ (2.5 mol%), **L5** (5.0 mol%), KOAc (2 equiv.), LiBr (30 mol%), ᵗBuOH/HFIP = 9:1, 100 °C for 12 h under N₂. Yields of the corresponding methyl esters after esterification with K₂CO₃ (2 equiv.) and MeI (5 equiv.) in NMP. [b]Isolated as the free acid. [c]1,4-dioxane instead of ᵗBuOH/HFIP = 9:1.

in the presence of KOAc to form the catalytically active monomeric species [Ru(**L6**)(OCOR)₂]. The catalytically active complex [Ru(**L6**)(OCOR)₂] first undergoes a reversible C−H activation to form *ortho*-carboxylate ruthenacycles **II**, in which the ligand increases the electron density of ruthenium center to facilitate the single-electron-reduction of **1a** to afford alkyl radical and a ruthenacycles intermediate **III**. Then, the alkyl radical then attacks the *para*-position of the C_Ar−Ru^III to form intermediate **VI**. In the next step, deprotonation with redox re-aromatization using KOAc produces intermediate **VII**, which undergoes protonation reactions to release the product **3aa** and regenerate the catalyst [Ru(**L6**)(OCOR)₂]. The Lewis acid LiBr helps to relocate the substrate, breaking the bridging coordination of the potassium and facilitating the release of product **3aa** from **VII** via protodemetalation.

### Applications
To illustrate the synthetic utility of the *meta*-C−H alkylated of aromatic carboxylic acid products, we performed various follow-up transformations of **3ba**. As can be seen from Fig. 5, the carboxylate group can be reused as a directing group for *ortho*-C−H functionalization, such as arylations (**6a**)[51], alkenylation/C−O cyclizations

(**6b**)[52], alkenylation/decarboxylations (**6c**)[24], further increasing the complexity of the structures. The carboxylate group can also be removed tracelessly (**6d**)[53] or serve as leaving group in a decarbonylative cyanation (**6e**)[54]. Furthermore, the carboxylate was reduced to the corresponding benzyl alcohol (**6f**)[55] and to the aldehyde (**6g**)[56]. The carboxylate group was also transformed into a difluoromethyl ester (**6h**)[57], a ketone (**6i**)[58] and an amide (**6j**)[59]. These examples show the advantages of using such a versatile group as a carboxylic acid as a directing group in *meta*-C−H alkylations. In contrast to conventional directing groups, it opens up vast opportunities for follow-up reactions and can also be easily removed.

### Discussion
A catalyst system generated from [RuCl₂(p-cym)]₂ and 5,5′-dimethylbipyridine was found to enable the *meta*-alkylation of aromatic carboxylic acids with secondary and tertiary alkyl bromides bearing various functional groups. Remarkably, 1-naphthoic acids direct the alkylation selectively into the C5-position, i.e, into the second aromatic ring. The reaction concept was shown to be broadly applicable to the synthesis of diversely functionalized arenes in substitution

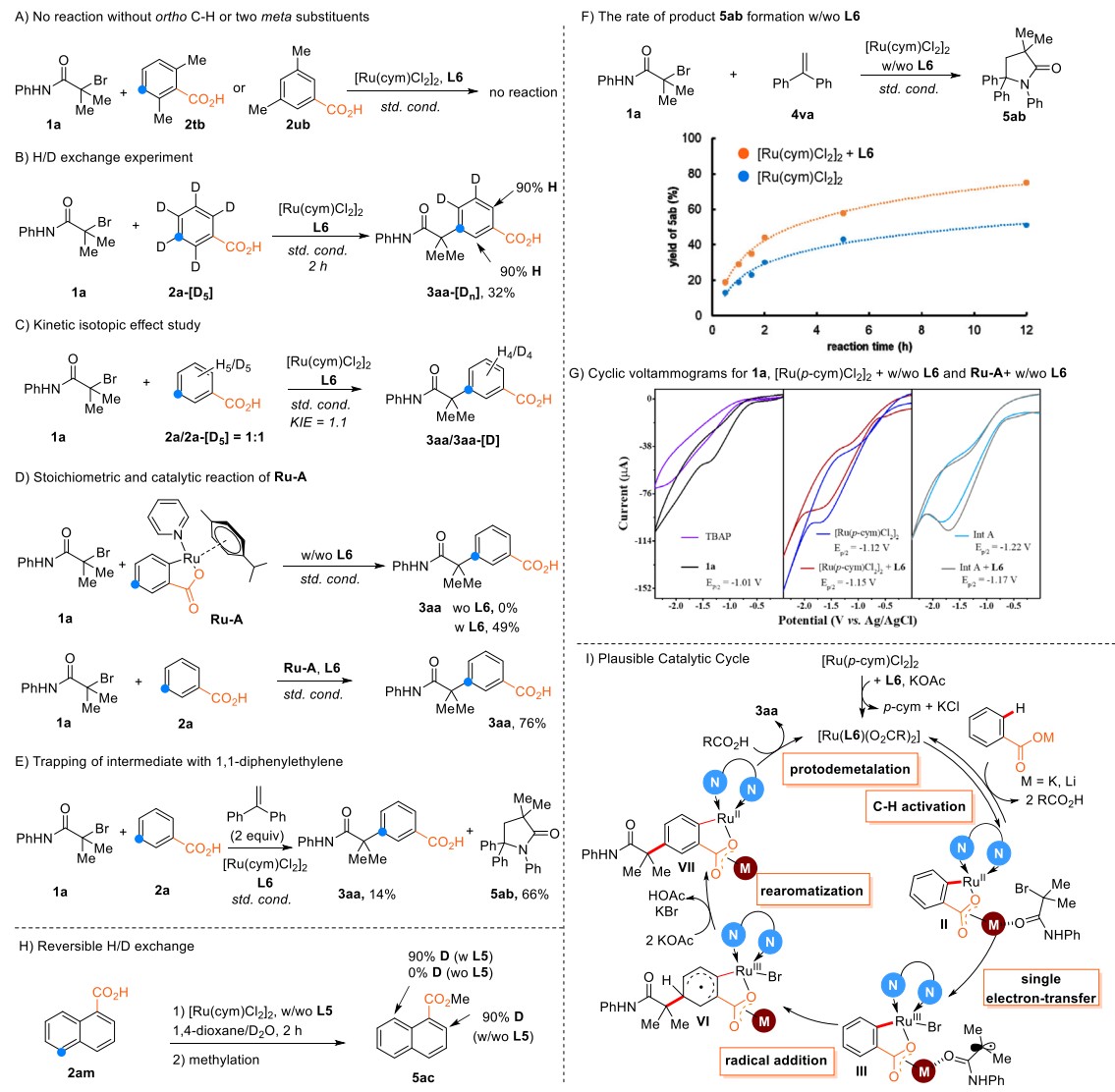

**Fig. 4 | Mechanistic studies. A** No reaction without *ortho* C–H or two *meta* substituents. **B** H/D exchange experiment. **C** Kinetic isotopic effect study. **D** Stoichiometric and catalytic reaction of **Ru-A**. **E** Trapping of intermediate with 1,1-diphenylethylene. **F** The rate of product **5ab** formation w/wo **L6**. **G** Cyclic voltammograms for **1a**, [Ru(*p*-cym)Cl₂]₂ + w/wo **L6** and **Ru-A** + w/wo **L6**. **H** Reversible H/D exchange. **I** Plausible catalytic cycle.

patterns that are not easily accessible otherwise. The carboxylate directing group is an ideal anchor point for follow-up functionalization, as demonstrated by the diversification of the products via various follow-up reaction. This makes the reaction ideal for late-state functionalizations of natural products or synthetic drugs.

Control experiments revealed that the reaction proceeds via rapid and reversible *ortho*-C–H metalation to give *ortho*-benzoate ruthenacycles, which reduce the alkyl halides to the corresponding radicals. These attack the ruthenacycles at the C–H bond *para* to the C$_{Ar}$–Ru bond allowing for a regiospecific synthesis of diversely functionalized C–H alkylated products from easily accessible starting materials. Adjusting the proton activity and basicity of the solvent system proved to be the decisive factor for achieving selectivity for C–H alkylation over intramolecular side reactions of the alkyl halides. Control experiments revealed the crucial role of potassium bases and the beneficial effect of Lewis acidic additives. They also revealed that bipyridine ligand accelerates the reversible cyclometallation step and facilitates the generation of the alkyl radicals by increasing the redox potential of the ruthenacycles.

## Methods

### Representative procedure for ruthenium catalyzed *meta*-C−H alkylation of (hetero)aromatic carboxylic acids

The procedure was conducted in a nitrogen-filled glove box. To a reaction vial equipped with a magnetic stir bar was added alkyl bromide **1** (0.3 mmol, 1.0 equiv.), aromatic carboxylic acids **2** (0.6 mmol, 2.0 equiv.), [Ru(*p*-cym)Cl₂]₂ (4.6 mg, 2.5 mol%), 5,5'-di-Me-bpy (2.8 mg, 5 mol%), KOAc (58.8 mg, 2.0 equiv.), LiBr (7.8 mg, 30 mol%), *t*BuOH (1.8 mL) and HFIP (0.2 mL). The reaction vial was sealed and removed from the glove box. The resulting mixture was stirred at 100 °C for 12 h, then quenched with HCl (1 M, 100 mL) and extracted with ethyl acetate (100 mL). The combined organic layers were dried over anhydrous MgSO₄ and concentrated under reduced pressure. The residue was purified by flash column chromatography (petroleum ether/ethyl acetate/formic acid = 80/20/1) to give the corresponding products or the crude product was isolated as the corresponding methylation product to which K₂CO₃ (2 equiv.), MeI (5 equiv.) and NMP (2.0 mL) were added at 60 °C for a further 2 h. The mixture was then quenched with saturated sodium chloride solution (50 mL) and extracted with ethyl acetate (100 mL). The combined organic layers were dried over

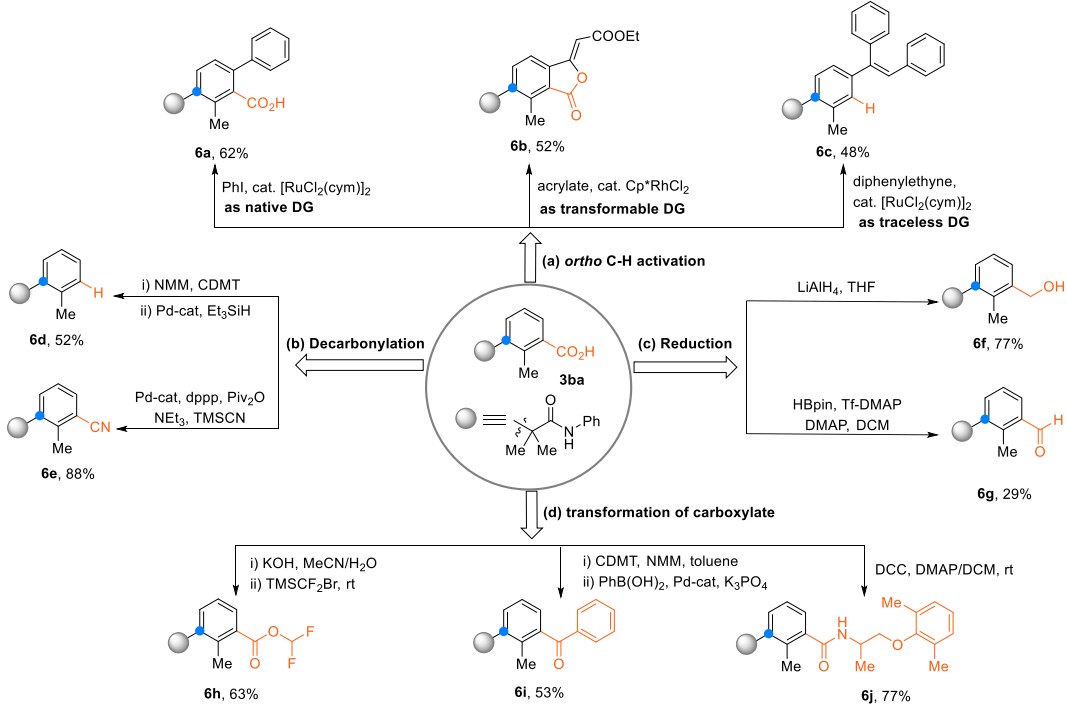

**Fig. 5 | Synthetic applications. a** Carboxylate directed *ortho*-C–H functionalization. **b** Decarbonylation of ArCO₂H. **c** Reduction of ArCO₂H. **d** Transformation of ArCO₂H.

anhydrous MgSO₄ and concentrated under reduced pressure. The residue was purified by column chromatography (petroleum ether/ethyl acetate) to give the alkylation product in the form of its methyl ester.

## General procedure for ruthenium catalyzed C5-alkylation of 1-naphthoic acids

The procedure was conducted in a nitrogen-filled glove box. To a reaction vial equipped with a magnetic stir bar was added alkyl bromide **1** (0.3 mmol, 1.0 equiv.), aromatic carboxylic acids **2** (0.6 mmol, 2.0 equiv.), [Ru(*p*-cym)Cl₂]₂ (4.6 mg, 2.5 mol%), 4,4'-di-CF₃-bpy (4.4 mg, 5 mol%), KOAc (58.8 mg, 2.0 equiv.), LiBr (7.8 mg, 30 mol%), ᵗBuOH (1.8 mL) and HFIP (0.2 mL). The reaction vial was sealed and removed from the glove box. The resulting mixture was stirred at 100 °C for 12 h, then quenched with HCl (1 M, 100 mL) and extracted with ethyl acetate (100 mL). The combined organic layers were dried over anhydrous MgSO₄ and concentrated under reduced pressure. The residue was purified by flash column chromatography (petroleum ether/ethyl acetate/formic acid = 80/20/1) to give the corresponding products or the crude product was isolated as the corresponding methylation product to which K₂CO₃ (2 equiv.), MeI (5 equiv.) and NMP (2.0 mL) were added at 60 °C for a further 2 h. The mixture was then quenched with saturated sodium chloride solution (50 mL) and extracted with ethyl acetate (100 mL). The combined organic layers were dried over anhydrous MgSO₄ and concentrated under reduced pressure. The residue was purified by column chromatography (petroleum ether/ethyl acetate) to give the alkylation product in the form of its methyl ester.

## Data availability

The authors declare that the data relating to the materials and methods, experimental procedures, NMR spectrum, mechanistic studies and synthetic application and X-ray structural analysis are available within this manuscript and in the Supplementary Information file. The X-ray crystallographic coordinates for structures **3ba**, **3kb** and **4aa** reported in this study have been deposited at the Cambridge Crystallographic Data Centre (CCDC) under CCDC 2245724, CCDC 2284237 and CCDC 2245725, respectively. These data are available free of charge from the Cambridge Crystallographic Data Centre via www.ccdc.cam.ac.uk/data_request/cif. Data is also available from the authors upon request.

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

## Acknowledgements

The authors thank National Natural Science Foundation of China (No. 21971074, L.H.; 22001076, L.H.), Natural Science Foundation of Guangdong Province (No. 2022A1515010660, L.H.; 2021A1515220024, L.H.) and the Deutsche Forschungsgemeinschaft (DFG, German Research Foundation) under Germany's Excellence Strategy–EXC 2033-390677874-RESOLV (L.-J.G.) for financial support.

## Author contributions

X.L. and L.H. conceived and designed the experiments. X.L performed experiments and wrote the paper. Both L.H. and L.-J.G. revised reviewed and edited the paper. P.H., J.S., Y.K., and F.S. carried out the experiments. H.J. contributed to discussions. All authors discussed the results and commented on the manuscript. L.H. and L.-J.G. directed the whole project.

## Funding

## Competing interests

The authors declare no competing interests.
