## [Peer Review File · Nature Communications]

REVIEWER COMMENTS

Reviewer #1 (Remarks to the Author):

The manuscript by Gooßen and Huang describes free carboxylic acid-directed meta-C-H alkylation of arenes enabled by a 2,2'-bipyridine ligand. Although ruthenium-catalyzed meta-C-H alkylation of arenes has comprehensively been investigated previously by several groups (Ackermann, Frost, Liang, et al.) using strongly coordination directing groups, the use of the weak coordination carboxylic acid as a directing group in this class of ruthenium-catalyzed remote meta-C-H functionalization is yet to be explored. The protocol demonstrates the wide compatibility with the different tertiary alkyl halides. Related to the common benzoic acids, site-selective C5-alkylation was successfully achieved for 1-naphthoic acids, which provide the useful 1,5-di-substituted naphthlene derivatives. The transformable carboxylic group makes the synthetic strategy far more interesting. Basically, this reviewer recommends its publishing after addressing the following concerns:

1. About optimization of reaction conditions (Table 1), it seems that Li cation plays an important role in the transformation (entry 15 versus entry 18). The author should provide a reasonable explain about this!
2. Owing to the investigated substrates (Table 2) do not include any heteroaromatic carboxylic acids, the manuscript title should be revised and just limited to aromatic carboxylic acids. Also in Abstract.

Reviewer #2 (Remarks to the Author):

Huang, Gooßen, and co-authors showed the regioselective alkylation of aromatic carboxylic acids with a wide range of substrate scope, highlighting its versatility and potential applicability to various synthetic targets. Formation of a key ruthenacycle intermediate is identified as crucial for directing the reaction towards meta-substitution with high regioselectivity and the presence of a bipyridyl ligand is critical for facilitating the SET process, which efficiently generates the alkyl radical necessary for the alkylation reaction. Compelling control experiments were conducted to support the proposed mechanism, strengthening the understanding of this intriguing reactivity. This method's potential to synthesize small molecules with applications in medicine and materials science, from new drugs to novel materials, makes it a strong fit for Nature Communications' focus on impactful science.

However, there are some suggestion and comments to this manuscript;

1) The authors propose that a potassium cation participates in the single-electron transfer (SET) process. Has a similar non-covalent interaction from a potassium cation been reported in previous literature? If so, please provide relevant references.

2) In Table 1, "yield of 3a" should be "yield of 3aa". And the authors should provide details on the methods used to quantify the byproducts.

3) For the scope of aromatic carboxylic acid in Table 2, we kindly suggest that the authors consider grouping each derivative as ortho-, meta-, and para-derivatives to help readers quickly identify trends in the reactivity of different isomers.

4) In Table 2, the authors state that the products were isolated as either free acids or methyl esters. For improved clarity, we suggest providing chemical structures of the isolated products as they are in the supporting information.

5) For clarity, Table 3 would benefit from including the numbering for 1-naphthoic acid, particularly on ruthenacycles IV and V. This would allow readers to easily follow the discussion and understand the specific positions involved in the reaction mechanism.

6) For clarity, we suggest the authors indicate the specific reaction times used in experiments B (Figure 2), which deviate from the standard conditions.

7) Although the authors mentioned the experiment with meta-disubstituent aromatic carboxylic acids in substrate scope part, the scheme of this reaction may be provided in Figure 2.

8) To enhance understanding of this chemistry, we suggest that the authors consider providing a proposed mechanism that outlines the key steps, role of catalysts, and formation of intermediates. This would be particularly helpful for readers who are less familiar with this specific reaction type.

Reviewer #3 (Remarks to the Author):

Huang and coworkers have described a very interesting reaction to introduce meta-alkyl substituents to aryl carboxylic acids by a ruthenium and bipyridine catalytic system. Several features make it worthy of publication in Nature Communications, like its efficiency, selectivity, employment of accessible feedstock, broad scope, and applications. They have also conducted several mechanistic experiments to propose a catalytic cycle.

In any case, some minor points require clarification before publication.

The highlighted advantage of this reaction “meta-C-H activation of various ArCO₂H”. In the reaction, the C-H activation happened at the ortho position of the carboxylate group, thus it is better to change the description.

The claimed “noncovalent interaction enabled reaction” is not demonstrated directly either experimentally or theoretically.

In the third paragraph, “1) As carboxylates are hard O-nucleophiles, they transform only slowly into metallacycles II40-42.” In the references, only palladium-catalysts were cited, and the carboxylate directed cyclometallation with ruthenium are common and C-H activation (cyclometallation) usually is not the rate-determining step.

Dear Editor Enda Bergin,

We thank the reviewers for their insightful comments, which helped us to further improve the quality of our manuscript. We have carefully revised our manuscript to address the points raised by the reviewers, as detailed below. We have also attached a manuscript file with all revisions highlighted in yellow.

Reviewer #1 (Remarks to the Author):

Comments: The manuscript by Gooßen and Huang describes free carboxylic acid-directed meta-C-H alkylation of arenes enabled by a 2,2'-bipyridine ligand. Although ruthenium-catalyzed meta-C-H alkylation of arenes has comprehensively been investigated previously by several groups (Ackermann, Frost, Liang, et al.) using strongly coordination directing groups, the use of the weak coordination carboxylic acid as a directing group in this class of ruthenium-catalyzed remote meta-C-H functionalization is yet to be explored. The protocol demonstrates the wide compatibility with the different tertiary alkyl halides. Related to the common benzoic acids, site-selective C5-alkylation was successfully achieved for 1-naphthoic acids, which provide the useful 1,5-di-substituted naphthlene derivatives. The transformable carboxylic group makes the synthetic strategy far more interesting. Basically, this reviewer recommends its publishing after addressing the following concerns:

Response: We thank the reviewer for his kind words.

Question 1: About optimization of reaction conditions (Table 1), it seems that Li cation plays an important role in the transformation (entry 15 versus entry 18). The author should provide a reasonable explain about this!

Response: This indeed an important point. We indeed did not include comparative experiments with other Lewis acids to prove that it is indeed the Lewis acidity of the Li-cation that plays a role. We have now added an entry to the screening Table 1, in which we show that Lewis acidic scandium triflate works as well as the lithium salts, and we have added a new table to the SI, in which we show the effect of several additives, among which only Lewis acids have a positive influence. We now comment on the effect of the lithium as follows:

The high efficiency of a weak base and a proton-active solvent suggested that the release of product **3aa** from **III** via protodemetalation and salt metathesis with **2a** might still be sluggish. We, thus, added Lewis acids to facilitate these steps. Indeed, the presence of lithium bromide markedly improved the conversion without negatively affecting the selectivity (entries 15-18). This effect can be assigned to the Lewis acidic cation rather than the counter ion, because neither the addition of excess KBr nor the removal of bromide by silver salts has a decisive effect, whereas other Lewis acids such as Sc(OTf)₃ were also beneficial (entries 17-19 and Table S4).

Table S4: Influence on additives on the reaction efficiency demonstrating the beneficial effect of Lewis acidic cations. ^[a]

	+		$\xrightarrow[\text{additive (30 mol\%)}]{\text{Ru}(p\text{-cym})\text{Cl}_2]_2 \text{ (5 mol\%)}$ $\text{5,5'-di-Me-bpy (10 mol\%)}$ KOAc (2 equiv.) $\text{tBuOH:HFIP = 9:1, 100 }^\circ\text{C}$ then methylation steps	Entry	Additive	3aa /(yield)% ^b		
1	-	61		
2	LiOTf	78		
3	KOTf	60		
4	LiOAc	84		

5	KOAc	51
6	LiCl	81
7	KCl	57
8	LiBr	86
9	KBr	51
10	Sc(OTf)₃	85
11	MgBr₂	66

^[a] Reaction conditions: **1a** (0.1 mmol), **2a** (0.2 mmol), [Ru(*p*-cym)Cl₂]₂ (5 mol%), 5,5-di-Me-bpy (10 mol%), KOAc (2 equiv.), additive (30 mol%), ^tBuOH:HFIP = 9:1 (1 mL), 100 ° C, N₂, 12 h. ^b The yields of the corresponding methyl esters were determined by GC analysis after esterification with K₂CO₃ (2.0 equiv.) and MeI (5.0 equiv.) in NMP using *n*-tetradecane as internal standard.

Question 2: Owing to the investigated substrates (Table 2) do not include any heteroaromatic carboxylic acids, the manuscript title should be revised and just limited to aromatic carboxylic acids. Also in Abstract.

Response: We are a bit confused by this comment since the substrates in Table 2 do include heteroaromatic carboxylic acids. 1-methylindole (**3ab**), benzofuran (**3ac**) and benzothiophene (**3ad**) clearly qualify as heterocycles. However, the reviewer probably wants to express his concern that some classical heterocycles such as pyridines are absent from the scope table. We have added the following sentences to the main text and the abstract to clarify remaining limitations. However, changing the title would be difficult, as we cannot specify within a single line which heterocycles work and which ones do not.

Abstract:

The resulting catalytic *meta*-C–H alkylation extends to a wide range of (hetero)aromatic carboxylic acids including benzofused five-membered ring heteroarenes but no pyridine derivatives in combination with secondary/tertiary alkyl halides, including fluorinated derivatives.

Main text:

The reaction extends to benzofused carboxylates and bicyclic heterocycles such as naphthalene (**3wa**), benzodioxane (**3xa-za**), 1-methylindole (**3ab**), benzofuran (**3ac**) and benzothiophene (**3ad**) but not yet to five- or six membered heteroarene carboxylates. Unfortunately, strongly coordinating 6-membered heterocycles such as pyridines or pyrimidines are not tolerated.

Reviewer #2 (Remarks to the Author):

Comments: Huang, Gooßen, and co-authors showed the regioselective alkylation of aromatic carboxylic acids with a wide range of substrate scope, highlighting its versatility and potential applicability to various synthetic targets. Formation of a key ruthenacycle intermediate is identified as crucial for directing the reaction towards *meta*-substitution with high regioselectivity and the presence of a bipyridyl ligand is critical for facilitating the SET process, which efficiently generates the alkyl radical necessary for the alkylation reaction. Compelling control experiments were conducted to support the proposed mechanism, strengthening the understanding of this intriguing reactivity. This method's potential to synthesize small molecules with applications in medicine and materials science, from new drugs to novel materials, makes it a strong fit for Nature Communications' focus on impactful science.

Response: We thank the reviewer for his kind words.

However, there are some suggestion and comments to this manuscript;

Question 3: The authors propose that a potassium cation participates in the single-electron transfer (SET) process. Has a similar non-covalent interaction from a potassium cation been reported in previous literature? If so, please provide relevant references.

Response: Thanks for your kind advice. There are reactions in the literature (Angew. Chem. Int. Ed. 2018, 57, 15762–15766 and J. Am. Chem. Soc. 2017, 139, 7745–7748) in which the presence of non-covalent interactions with bridging potassium ions are vital to explain the regioselectivity of C-H boronations (as below).

Based on these finding, we found it safe to assume that both substrates of our reaction are also bridged by non-covalent interactions with metal cations. However, in our case, they are not required to explain the regioselective outcome, and there is no spectroscopic or other direct evidence to pinpoint the existence of non-covalent interactions for this transformation. In response to the criticism of two reviewers (see also below), we removed the explicit reference to non-covalent bonding effects from our mechanistic sketch and no longer present them as being enabled by non-covalent interactions. The modifications were made:

Main text:

Introduction

We hypothesized that a catalytic concept for the so far elusive meta-C–H alkylation of aromatic carboxylate must address these issues in the following way: 1) The tendency of the alkyl halides to undergo an S_N2 reaction with carboxylates must be reduced. 2) Electron-rich ligands must be added to increase the electron density of the resulting ruthenacycles, thus increasing the reduction potential of Int. II⁴⁵ and the C–H reactivity of Int. II (Fig. 1c). 3) Non-covalent interactions between the carboxylate group and functional groups in the alkyl halides must be utilized to bring the reactants into closer proximity⁴⁶⁻⁴⁷, thereby facilitating the single electron reduction and the radical addition steps.

Optimization conditions section

This indicates that solvent-stabilized, potassium-bridged assemblies of the two substrates (II, III) are involved in the selectivity-determining steps. The high efficiency of a weak base and a proton-active solvent suggested that the release of product 3aa from III via protodemetalation and salt metathesis with 2a might still be sluggish.

Figure 1.

c) This work: ligand enabled *meta*-C-H alkylation of ArCO₂H with alkyl halides

Question 4: In Table 1, “yield of 3a” should be “yield of 3aa”. And the authors should provide details on the methods used to quantify the byproducts.

Response: Thanks for your advice. We have corrected the mistake “yield of 3a” to “yield of 3aa” in Table 1 of the revised manuscript. We have used gas chromatography to quantify the yields of by-products 4a, 5a and 6a using *n*-tetradecane as the internal standard. This is now stated in the revised manuscript as follows:

Table 1.

entry ^c	additive ^c	base ^c	ligand ^c	solvent ^c	yield of 3aa [%] ^{b,c}	byproducts 4a/5a/6a [%] ^c

^b Yields of the corresponding methyl esters after esterification with K₂CO₃ (2 equiv.) and MeI (5 equiv.), and isolated yield in parentheses.

^c Yields were determined using GC yields with *n*-tetradecane as internal standard.

Question 5: For the scope of aromatic carboxylic acid in Table 2, we kindly suggest that the authors consider grouping each derivative as *ortho*-, *meta*-, and *para*-derivatives to help readers quickly identify trends in the reactivity of different isomers.

Response: Thanks for your kind advice. We have now grouped the substrates according to their substitution pattern in Table 2, and discuss the scope in this order as follows:

Table 2. The scope of the *meta*-alkylation of aromatic carboxylic acids.^[a]

Main text:

Many *ortho*-substituted aromatic carboxylic acid were selectively converted into 1,2,3-trisubstituted arenes. The preference for these thermodynamically less favorable products indicates the sensitivity of the *ortho*-metalation step towards steric hindrance. Alkyl (**3ba-ca**), alkoxy (**3da**), fluoro (**3ea**) and chloro (**3fa**) substituents were all tolerated whereas - in analogy to related processes - nucleophilic hydroxyl and amino groups were found to be incompatible. The directed alkylation of *meta*-substituted aromatic carboxylic acids delivers 1,3,5-trisubstituted aromatic carboxylic acids (**3ga-ia**). Benzoates bearing *para*-substituents were also smoothly converted (**3ja-pa**), which shows that in contrast to the metalation, the radical addition step is not hampered by steric hindrance. Multi-substituted aromatic acids bearing various functional groups were also successfully converted into the corresponding products (**3qa-va**).

Question 6: In Table 2, the authors state that the products were isolated as either free acids or methyl esters. For improved clarity, we suggest providing chemical structures of the isolated products as they are in the supporting information.

Response: Thanks for your advice. We have now provided the chemical structures of the isolated products in the revised manuscript in Table 2 and Table 3 as follows:

Scope of aromatic carboxylic acid

Scope of alkyl bromide

Difluoroalkyl bromide

Late-stage functionalization of drug molecule

Question 7: For clarity, Table 3 would benefit from including the numbering for 1-naphthoic acid, particularly on ruthenacycles IV and V. This would allow readers to easily follow the discussion and understand the specific positions involved in the reaction mechanism.

Response: Thanks for your advice. In the revised manuscript, we renumbered the 1-naphthoic acids including the ruthenacycles intermediates IV and V. The passage now reads as follows.

Question 8: For clarity, we suggest the authors indicate the specific reaction times used in

experiments B (Figure 2), which deviate from the standard conditions.

Response: This is a valid point. We have corrected the reaction times in experiments B (Figure 2) in the revised manuscript as shown below.

B) H/D exchange experiment

Question 9: Although the authors mentioned the experiment with meta-disubstituent aromatic carboxylic acids in substrate scope part, the scheme of this reaction may be provided in Figure 2.

Response: We appreciate this suggestion. In the revised manuscript, the reaction with meta-disubstituent aromatic carboxylic acids (2ub) is depicted in Figure 2A and is discussed as follows:

A) No reaction without *ortho* C-H or two *meta* substituents

Main text:

A) The attempted reaction of 2,6-dimethylbenzoic acid (**2tb**) or aromatic carboxylic acids with two *meta*-methyl substituents (**2ub**) gave no conversion, which confirms that the presence of *ortho* C-H bonds is vital for the *meta*-functionalization to proceed.

Question 10: To enhance understanding of this chemistry, we suggest that the authors consider providing a proposed mechanism that outlines the key steps, role of catalysts, and formation of intermediates. This would be particularly helpful for readers who are less familiar with this specific reaction type.

Response: Thanks for your kind advice. We have added the mechanistic cycle in the revised manuscript. The main step of this transformation includes: C-H activation, SET reduction of 1a, radical addition, aromatization and protodemetalation.

Main text:

1) Based on these control experiments, a plausible mechanism for *meta*-C–H alkylation of aromatic carboxylic acids is proposed. Initially, $[\text{Ru}(\textit{p}\text{-cym})\text{Cl}_2]_2$ reacts with **L6** in the presence of KOAc to form the catalytically active monomeric species $[\text{Ru}(\text{L6})(\text{OCOR})_2]$. The catalytically active complex $[\text{Ru}(\text{L6})(\text{OCOR})_2]$ first undergoes a reversible C–H activation to form *ortho*-carboxylate ruthenacycles **II**, in which the ligand increases the electron-density of ruthenium center to facilitate the single-electron-reduction of **1a** to afford alkyl radical and a ruthenacycles intermediate **III**. Then, the alkyl radical then attacks the *para*-position of the $\text{C}_{\text{Ar}}\text{--Ru}^{\text{III}}$ to form intermediate **VI**. In the next step, deprotonation with redox re-aromatization using KOAc produces intermediate **VII**, which undergoes protonation reactions to release the product **3aa** and regenerate the catalyst $[\text{Ru}(\text{L6})(\text{OCOR})_2]$. The Lewis acid LiBr helps to relocate the substrate, breaking the bridging coordination of the potassium and facilitating the release of product **3aa** from **III** via protodemetalation.

Reviewer #3 (Remarks to the Author):

Comments: Huang and coworkers have described a very interesting reaction to introduce meta-alkyl substituents to aryl carboxylic acids by a ruthenium and bipyridine catalytic system. Several features make it worthy of publication in Nature Communications, like its efficiency, selectivity, employment of accessible feedstock, broad scope, and applications. They have also conducted several mechanistic experiments to propose a catalytic cycle.

In any case, some minor points require clarification before publication.

Response: We thank the reviewer for his kind words.

Question 11: The highlighted advantage of this reaction “*meta*-C-H activation of various ArCO_2H ”. In the reaction, the C-H activation happened at the *ortho* position of the carboxylate group, thus it is better to change the description.

Response: The reviewer is correct, we should not speak about a *meta* C-H activation, because the activation is in *ortho*. We have now made sure that we instead consistently use the term *meta*-C-H functionalization, because this is what actually happens as the result of an *ortho*-C-H-activation. Thus, changes have been made to Page 1, lines 10 and 54 and Page 7, line 23.

Question 12: The claimed “noncovalent interaction enabled reaction” is not demonstrated directly either experimentally or theoretically.

Response: Thanks for your kind advice. As discussed above, there are literature reports (Angew. Chem. Int. Ed. 2018, 57, 15762–15766 and J. Am. Chem. Soc. 2017, 139, 7745–7748) in which non-covalent interactions between similar substrate combinations are postulated (see below). In this case, the presence of the non-covalent interactions is evident from the regioselectivity of the transformation. We cannot provide this kind of indirect evidence, and we do not have spectroscopic proof for the existence of non-covalent bonds although based on the below example, we believe that it is save to assume that we will see bridging coordination of cations.

Thus, we decided to remove the claim that the process is enabled by non-covalent bonding effects. The corrections are outlined above in our answer to reviewer 2.

Question 13: In the third paragraph, “1) As carboxylates are hard O-nucleophiles, they transform only slowly into metallacycles II 40-42.” In the references, only palladium-catalysts were cited, and the carboxylate directed cyclometallation with ruthenium are common and C-H activation (cyclometallation) usually is not the rate-determining step.

Response: Thanks for your kind advice. We have added two references to Ru catalyzed C-H activation as the rate-determining step (*Angew. Chem. Int. Ed.* 2016, 55, 6929–6932 and *Chem. Sci.*, 2018, 9, 5289–5294). In these two Ru-catalyzed C-H activation of aromatic carboxylic acids KIE results were respectively $K_H/k_D = 2.56$ and 3.0. The revision was as follows:

Main text:

- 1) As carboxylates are hard O-nucleophiles, they transform only slowly into metallacycles II^{22, 40-43}. This C–H activation step is rate-determining in most *ortho*-C–H functionalizations of carboxylates^{22, 44}.

We hope that you will find the manuscript acceptable for publication after these changes. If you have any further requests, we will be happy to address them.

Best regards,

Lukas Goßen and Lianbin Huang

REVIEWERS' COMMENTS

Reviewer #1 (Remarks to the Author):

The revised manuscript has addressed the concerns from the reviewers and is acceptable for publication.